# Adolescent Verbal Memory as a Psychosis Endophenotype: A Genome-Wide Association Study in an Ancestrally Diverse Sample

**DOI:** 10.3390/genes13010106

**Published:** 2022-01-03

**Authors:** Baihan Wang, Olga Giannakopoulou, Isabelle Austin-Zimmerman, Haritz Irizar, Jasmine Harju-Seppänen, Eirini Zartaloudi, Anjali Bhat, Andrew McQuillin, Karoline Kuchenbäcker, Elvira Bramon

**Affiliations:** 1Department of Mental Health Neuroscience, Division of Psychiatry, University College London, London W1T 7BN, UK; o.giannakopoulou@ucl.ac.uk (O.G.); isabelle.zimmerman.11@ucl.ac.uk (I.A.-Z.); a.irizar@ucl.ac.uk (H.I.); jasmine.harju-seppanen.16@ucl.ac.uk (J.H.-S.); e.zartaloudi@ucl.ac.uk (E.Z.); anjali.bhat.14@ucl.ac.uk (A.B.); a.mcquillin@ucl.ac.uk (A.M.); k.kuchenbaecker@ucl.ac.uk (K.K.); 2UCL Genetics Institute, Division of Biosciences, University College London, London WC1E 6BT, UK; 3Social, Genetic and Developmental Psychiatry Centre, Institute of Psychiatry, Psychology and Neuroscience, King’s College London, London SE5 8AF, UK; 4Department of Genetics & Genomic Sciences, Icahn School of Medicine at Mount Sinai, New York, NY 10029, USA

**Keywords:** verbal memory, genome-wide association study, psychosis, schizophrenia, endophenotype, neurodevelopment

## Abstract

Verbal memory impairment is one of the most prominent cognitive deficits in psychosis. However, few studies have investigated the genetic basis of verbal memory in a neurodevelopmental context, and most genome-wide association studies (GWASs) have been conducted in European-ancestry populations. We conducted a GWAS on verbal memory in a maximum of 11,017 participants aged 8.9 to 11.1 years in the Adolescent Brain Cognitive Development Study^®^, recruited from a diverse population in the United States. Verbal memory was assessed by the Rey Auditory Verbal Learning Test, which included three measures of verbal memory: immediate recall, short-delay recall, and long-delay recall. We adopted a mixed-model approach to perform a joint GWAS of all participants, adjusting for ancestral background and familial relatedness. The inclusion of participants from all ancestries increased the power of the GWAS. Two novel genome-wide significant associations were found for short-delay and long-delay recall verbal memory. In particular, one locus (rs9896243) associated with long-delay recall was mapped to the NSF (N-Ethylmaleimide Sensitive Factor, Vesicle Fusing ATPase) gene, indicating the role of membrane fusion in adolescent verbal memory. Based on the GWAS in the European subset, we estimated the SNP-heritability to be 15% to 29% for the three verbal memory traits. We found that verbal memory was genetically correlated with schizophrenia, providing further evidence supporting verbal memory as an endophenotype for psychosis.

## 1. Introduction

Psychotic disorders, including schizophrenia and bipolar disorder, are highly heritable (heritability estimated to be approximately 80%) [1,2]. Although the latest genome-wide association studies (GWASs) have discovered 270 genomic loci associated with schizophrenia and 64 associated with bipolar disorder [3,4], the mechanisms through which genetic variation in those loci contributes to disease risk are unknown. One potential solution to this issue is by investigating the endophenotypes of psychosis, which are biomarkers of the disease that lie closer to the genetic effects and are thought to be part of the mechanistic path from genetic variation to clinical manifestation [5,6]. Endophenotypes can offer insights into the biological mechanisms that are most pertinent to psychosis, thus informing our understanding of its complex aetiology.

Cognitive deficits of different severity are common in psychosis, and often manifest before symptom onset [7,8]. Therefore, psychosis has been suggested to be an outcome of neurodevelopmental abnormalities that occur in early life [9,10,11]. Among all cognitive domains, verbal memory impairment is one of the best characterised in psychosis. Previous meta-analyses have found prominent deficits in verbal memory in patients with schizophrenia (Cohen’s *d* = −1.20 to −0.85 [12]) and bipolar disorder (Cohen’s *d* = −0.56 to −0.50 [13]). Milder deficits have also been reported by meta-analyses on the relatives of patients with schizophrenia (Cohen’s *d* = −0.54 to −0.44 [14,15]) and bipolar disorder (Cohen’s *d* = −0.42 to −0.33 [16,17]). Those findings are consistent with verbal memory being associated with the genetic risk of psychosis, making it a potential endophenotype. Additionally, there is evidence that such deficits in verbal memory is already present in childhood and adolescence for those with increased genetic risk for psychosis [18,19]. Therefore, verbal memory impairment may reflect underlying neurodevelopmental abnormalities that convey the genetic risk of psychosis before its symptom onset or formal diagnosis.

Previous GWASs have found over 200 loci associated with general cognitive ability, as well as its negative genetic correlation with schizophrenia [20,21,22]. Chromosomal aberrations, including copy number variants, can also influence cognition [23,24,25]. Nevertheless, the population correlation between verbal memory and general cognitive ability at the phenotypic level was estimated to be 0.24 to 0.39, indicating that they are overlapping but different constructs [26]. Investigating the genetic basis of specific cognitive domains, such as verbal memory, could pinpoint the biological mechanisms that are most relevant to the aetiology of psychosis. However, there have been only a few GWASs conducted on verbal memory. The earliest findings came from Papassotiropoulos et al., who conducted GWASs in two samples from two Swiss cohorts (351 and 1073 participants) and found that delayed verbal episodic memory was associated with alleles in the KIBRA and CTNNBL1 genes [27,28]. Debette et al. conducted a GWAS meta-analysis in 29,076 older adults, and found that verbal declarative memory was associated with APOE, which has also been identified in GWASs on Alzheimer’s disease [29]. The association of verbal memory with APOE variants was replicated in a later GWAS by Arpawong et al. in 7486 older adults, with two additional significant SNPs identified in TOMM40 [30]. With regards to schizophrenia, in a sample of 127 patients and 136 controls, Nakahara et al. found that verbal memory was associated with a SNP near NDUFS4 [31]. Greenwood et al. conducted a GWAS on 11 endophenotypes for schizophrenia in a sample of 1533 participants, but found no genome-wide significant associations for verbal memory [32]. However, since the two largest GWASs on verbal memory were both conducted in older populations, their findings might be more relevant to neurodegenerative disorders such as Alzheimer’s disease instead of psychosis. Little is known about the biological basis of verbal memory in a neurodevelopmental context, and no GWASs on verbal memory have been conducted in children or young people so far.

The lack of diversity has long been an issue in GWASs, as 78% of published GWASs until 2019 were conducted only in European individuals [33]. Unsurprisingly, most previous GWASs on verbal memory have also been conducted in homogeneous European samples. This is mainly due to the lack of diversity in existing databases, as well as the statistical challenges of analysing samples from multiple ancestral backgrounds [34,35]. Such lack of diversity means we have a limited understanding of disease aetiology in underrepresented populations. By recruiting and including participants of all ancestries, we can maximise the power of GWASs, and may identify important genetic variants that are too rare to study in European populations [36]. One solution to these statistical difficulties is a meta-analysis of results of GWASs in multiple ancestries. However, this approach requires each of the primary GWASs in the meta-analysis to be conducted in a relatively homogeneous sample, and individuals of mixed ancestry are usually excluded from the analysis. Moreover, the sample size of each homogeneous ancestral group might be too small to estimate the small effects of individual variants. For heterogeneous multi-ancestry samples, joint mixed-model methods might be the optimal approach [34]. In the mixed models, modelling a genetic relationship matrix as a random effect allows the inclusion of individuals of mixed ancestry. This approach is not limited by the sample size of each homogeneous ancestral group and maximises statistical power [34,37,38]. One example of this is the linear mixed models with orthogonally partitioned structure included in the GENetic EStimation and Inference in Structured samples (GENESIS) R/Bioconductor package [37,38]. The GENESIS package partitions genetic structure into distant ancestral background and recent familial relatedness, thus accounting for ancestry better than other traditional mixed-model methods [37,38].

All in all, previous GWASs on verbal memory have mostly been conducted in European older adults, making them less relevant to psychosis as a neurodevelopmental condition. Therefore, the current study aims to investigate the genetic basis of verbal memory in adolescents as a neurodevelopmental endophenotype for psychosis. We conducted a GWAS on three verbal memory traits measured by the Adolescent Brain Cognitive Development (ABCD) Study^®^ in over 10,000 adolescents of diverse ancestries. We adopted a novel mixed-model approach using the GENESIS package to account for the diverse ancestries in the sample. We also estimated the heritability of the three verbal memory traits and their genetic correlations with other psychiatric traits, in order to investigate the shared genetic basis between verbal memory and psychosis.

## 2. Materials and Methods

### 2.1. Participants

The ABCD Study^®^ is a longitudinal study that recruited 11,878 participants aged about 9–11 years at baseline from 21 sites across the United States [39,40,41]. The 21 sites were nationally distributed and participants were recruited from schools in nearby catchment areas. Some schools were oversampled in order to obtain a sample representative of the national population, in terms of gender, ethnicity, socioeconomic status, and urbanicity [39]. Interested families were invited to the research sites and all children underwent a comprehensive set of demographic, physical, cognitive, mental health, and neuroimaging assessments, along with their blood and saliva samples collected for genotyping. Participants were excluded from the ABCD Study^®^ if they were not fluent in English, had a history of traumatic brain injury, or had a current diagnosis of schizophrenia, moderate/severe autism spectrum disorder, intellectual disability, or alcohol/substance use disorders [42,43]. For this GWAS, we used the phenotypic and genetic data at baseline from ABCD Data Release 3.0, which have now been anonymised and released to bona fide researchers after a registration process. Details on the recruitment procedures of the ABCD Study^®^ can be found in a previous publication [39].

### 2.2. Verbal Memory Assessment

The Rey Auditory Verbal Learning Test (RAVLT) is a verbal memory test widely used in research and clinical practice [44]. In the ABCD Study^®^, participants were firstly verbally presented with a list of 15 words (List A) and immediately asked to recall as many words from the list as possible. The same procedure was repeated four times for the same list (List A) after the first trial. The total number of words correctly recalled in the five trials was recorded as the RAVLT immediate recall score. After learning List A, participants were presented with a different list (List B, the interference list) and asked to recall words from List B. Right after recalling List B, participants were asked to recall List A again, and the correct number of words recalled in List A was recorded as the RAVLT short-delay recall score. Participants were assessed by other cognitive tests in the following 30 min, and were asked to recall list A at the end of the assessment. The final number of words correctly recalled from List A was recorded as the RAVLT long-delay recall score.

### 2.3. Genotyping, Imputation, and Quality Control

Participants were genotyped by the Rutgers University Cell and DNA Repository (RUCDR) using the Affymetrix NIDA SmokeScreen Array [45]. All genotyped data underwent standard quality control following the RICOPILI pipeline [46]. They were checked against the Haplotype Reference Consortium and the 1000 Genomes Project for consistencies [47,48], and then uploaded to the TOPMed server for imputation using mixed ancestry and Eagle v2.4 phasing [49,50]. We downloaded the imputed data from the ABCD Data Repository and used bcftools to annotate them with rs IDs based on dbSNP153 [51]. Imputed SNPs with dosage levels were converted to best-guess genotype format using PLINK v2.0 with a hard call threshold of 0.1 [52]. We performed further post-imputation quality control by removing SNPs with imputation quality score (*r*^2^) < 0.3 or minor allele frequency (MAF) < 0.01. The final sample included 11,017 participants and 11,229,083 variants after quality control. Details of genotyping, imputation, and quality control can be found in the Appendix A.

### 2.4. Relationship Inference and Principal Component Analysis

Since the ABCD Study^®^ included an ancestrally diverse sample with a combination of singletons, siblings, and twins, we used the GENESIS package in R-4.0.3 to conduct a linear mixed-model GWAS [38,53]. Based on the genotyped data after quality control, we firstly used KING-robust 2.2.5 to infer the unadjusted kinship coefficient for each pair of participants in the dataset [54]. We then used the PC-AiR package [38] and the output from KING-robust to conduct a principal component analysis of ancestral background that accounts for familial relatedness. This was performed on the genotyped data after pruning using the SNPRelate package in R [55]. Finally, to get a more accurate estimation of familial relatedness, we performed a PC-Relate analysis to generate a new kinship matrix with kinship coefficients adjusted for ancestral background [56]. To perform an ancestry-specific analysis in comparison to the multi-ancestry analysis, we also identified a subset of participants who were ancestrally European based on the principal component analysis. The same procedure was repeated to estimate the principal components and familial relatedness within the European subset. However, we were unable to analyse participants of other ancestries due to their small sample sizes. Details of relationship inference and principal component analysis can be found in the Appendix A.

### 2.5. Genome-Wide Association Testing

A mixed-model genome-wide association test was conducted using the GENESIS package for the overall sample and the European subset [37]. The RAVLT immediate recall score, short-delay recall score, or long-delay recall score was added as an outcome. Age, sex, the first eight (for the overall sample) or four (for the European subset) ancestrally representative principal components were included as fixed effects, and the ancestry-adjusted kinship matrix was included as a random effect. The number of principal components was chosen based on visual inspection of the principal component analysis plots. Principal components were included in the model if they showed variability across ancestries. The genome-wide significant p-value threshold was set to 5 × 10^−8^. Additionally, we checked if the lead SNPs in loci that reached genome-wide significance for one verbal memory trait also reached the nominal significance level for the other two traits.

### 2.6. Locus Definition and Gene Mapping

Summary statistics of the GWAS results of the three verbal memory traits in the whole sample were uploaded to the platform of Functional Mapping and Annotation of Genome-Wide Association Studies (FUMA) [57]. A reference panel from all populations in the 1000 Genomes Project (Phase 3) was used to obtain *r*^2^ [47]. In FUMA, independent significant SNPs were firstly defined as SNPs with *p* < 5 × 10^−8^ independent from each other at *r*^2^ < 0.6. Lead SNPs of genomic loci were then defined as a subset of independent significant SNPs at *r*^2^ < 0.1. Lead SNPs with a distance closer than 250 kb were merged to one locus. SNPs with *r*^2^≥ 0.6 were considered to be in linkage disequilibrium (LD) with the lead SNPs and thus their good proxies.

We used three strategies to map the lead SNPs and their good proxies to protein-coding genes. (1) Positional mapping: SNPs were mapped to genes if the SNPs are located within a 10 kb window around the gene boundaries. (2) Expression quantitative trait loci (eQTL) mapping: SNPs were mapped to genes if the SNPs have a significant impact on the gene expression. eQTLs were selected from GTEx v8 brain tissues and only significant SNP-gene pairs in GTEx (FDR < 0.05) were used [58]. Mapping was based on cis-eQTLs with a 1Mb window around each gene. (3) Chromatin interaction mapping: SNPs were mapped to genes if there are significant chromatin interactions between the SNPs and nearby or distant genes. Chromatin interaction information of four tissues was obtained from Hi-C, including adult cortex, fetal cortex, dorsolateral prefrontal cortex, and hippocampus [59,60]. Only significant interactions were used based on an FDR cutoff of 10^−6^. Promoter regions were defined as 250 bp upstream and 500 bp downstream from the transcriptional start site. SNPs were filtered by enhancers and genes were filtered by promoters based on annotations from the Roadmap Epigenomics Project brain tissue data [61].

### 2.7. Heritability and Genetic Correlations

Based on the GWAS results of the European subset in the ABCD Study^®^, we used linkage disequilibrium score regression (LDSC) to estimate the SNP heritability (*h^2^*; phenotypic variance explained by common SNPs) of the three verbal memory traits. We also used LDSC to test their genetic correlations (*r_g_*) with each other, as well as with three psychiatric traits (schizophrenia, bipolar disorder, and major depressive disorder) and educational attainment. LD scores calculated from the European participants in the 1000 Genomes Project (Phase 3) were used [47]. We used the latest GWAS summary statistics of schizophrenia, bipolar disorder, and major depressive disorder from the Psychiatric Genomics Consortium (PGC; https://www.med.unc.edu/pgc/; accessed on 2 November 2021) [3,4,62]. The GWAS summary statistics of major depressive disorder excluded the 23andMe sample due to the data transfer agreement between the PGC and 23andMe. The GWAS summary statistics of educational attainment were retrieved from the Social Science Genetic Association Consortium (https://www.thessgac.org/; accessed on 5 October 2021) [63]. All GWAS summary statistics included only participants of European ancestries.

## 3. Results

### 3.1. Sample Characteristics

As shown in Table 1, among the 11,017 participants who passed genetic quality control in the ABCD Study^®^, there were 5176 females (47%) and 5805 males (53%). The mean age of the participants was 9.9 years (SD = 0.6 years, range = 8.9 to 11.1 years). Based on broadly defined ethnicity reported by parents, 5872 (53%) participants were reported as White, 2099 (19%) as Hispanic, 1684 (14%) as Black, and 1% as Asian. The remaining 11% included children of American Indian/Alaska Native, Native Hawaiian and other Pacific Islander, and mixed ethnicity. Participants scored on average 44.2 for RAVLT immediate recall (SD = 9.9), 9.7 for RAVLT short-delay recall (SD = 3.0), and 9.2 for RAVLT long-delay recall (SD = 3.2). The three verbal memory traits were highly correlated at the phenotype level: *r* = 0.72 between RAVLT immediate recall and short-delay recall (*p* < 0.001); *r* = 0.73 between RAVLT immediate recall and long-delay recall (*p* < 0.001); *r* = 0.79 between RAVLT short-delay recall and long-delay recall (*p* < 0.001). All three verbal memory traits were approximately normally distributed in the sample (Appendix A).

### 3.2. Principal Component Analysis and Relatedness Estimation

32 principal components were returned by PC-AiR for the whole dataset. The distribution of the samples along the first eight principal components coloured by reported ethnicity can be seen in Appendix A. Based on the plots of the principal components and participants’ ethnicity reported by their parents, we identified a subset of 5763 participants of European ancestry (Appendix A). In the whole sample, PC-Relate revealed 416 pairs of monozygotic twins (kinship coefficient > 0.354) and 1356 pairs of dizygotic twins or siblings (0.176 < kinship coefficient ≤ 0.354; Appendix A).

### 3.3. Genome-Wide Association Testing

The GWAS in the European subset included 5635 participants for RAVLT immediate recall, 5667 participants for RAVLT short-delay recall, and 5640 participants for RAVLT long-delay recall. No genome-wide significant associations were found for any of the three verbal memory traits (Appendix A). In contrast, the inclusion of non-European participants increased the sample sizes to 10,726 participants for RAVLT immediate recall, 10,801 participants for RAVLT short-delay recall, and 10,757 participants for RAVLT long-delay recall. The GWAS results in the whole sample are shown in Figure 1. No genome-wide significant SNPs were identified for RAVLT immediate recall. For RAVLT short-delay recall, one SNP (rs73984566) in an intergenic region of chromosome 2 reached genome-wide significance (beta = −0.847, standard error (SE) = 0.151, *p* = 1.86 × 10^−8^). For RAVLT long-delay recall, we found one genome-wide significant SNP (rs9896243) located in the intron of the NSF gene in chromosome 17 (beta = −0.309, SE = 0.055, *p* = 2.22 × 10^−8^). Table 2 shows the detailed information about each genome-wide significant locus. Both genome-wide significant SNPs associated with one verbal memory trait also reached the nominal significance threshold for the other two verbal memory traits, with an effect size in the same direction (Appendix A).

### 3.4. Gene Mapping

For the gene mapping we considered the lead SNPs in the two novel loci, as well as their good proxies. In the locus associated with RAVLT short-delay recall (lead SNP: rs73984566), there was strong evidence for two genes, NABP1 and TMEFF2, based on chromatin interaction data in fetal cortex. The lead SNP (rs9896243) in the locus associated with RAVLT long-delay recall is located in the intron of the NSF gene, which was supported by positional mapping of the locus in FUMA. Additionally, a good proxy of the lead SNP (rs199533; *r*^2^ = 0.61) results in a synonymous amino acid change in this gene according to HaploReg v4.1 [64]. This SNP (rs199533) was genotyped in the study and showed an association with RAVLT long-delay recall at the nominal significance level (beta = −0.285, *p* = 1.76 × 10^−6^). Other variants in the locus have also been shown to affect the expression of the same gene in brain tissues. Additionally, the locus was mapped to seven other genes in the locus (17q21.31) based on brain eQTLs. Details of all mapped genes are summarised in Appendix A.

### 3.5. Heritability and Genetic Correlations

Based on the GWAS in the European subset, LDSC estimated the SNP heritability (*h^2^*) to be 0.15 (SE = 0.07) for RAVLT immediate recall, 0.29 (SE = 0.07) for RAVLT short-delay recall, and 0.21 (SE = 0.07) for RAVLT long-delay recall. The genetic correlations among the three verbal memory traits were very high: *r_g_* = 0.99 (SE = 0.12) between RAVLT immediate recall and short-delay recall (*p* < 0.001); *r_g_* = 0.91 (SE = 0.12) between RAVLT immediate recall and long-delay recall (*p* < 0.001); *r_g_* = 0.93 (SE = 0.06) between RAVLT short-delay recall and long-delay recall (*p* < 0.001). The results of their genetic correlations with other psychiatric traits are shown in Table 3. All three verbal memory traits were significantly and positively correlated with educational attainment (*r_g_* = 0.31–0.53, *p* ≤ 0.001). Schizophrenia was found to be significantly and negatively correlated with both RAVLT immediate recall (*r_g_* = −0.29, SE = 0.10, *p* = 0.003) and RAVLT short-delay recall (*r_g_* = −0.15, SE = 0.06, *p* = 0.017), as well as RAVLT long-delay recall at a trend level (*r_g_* = −0.15, SE = 0.08, *p* = 0.063). We also found a significant negative genetic correlation between RAVLT short-delay recall and major depressive disorder (*r_g_* = −0.14, SE = 0.07, *p* = 0.050).

## 4. Discussion

This is one of the first studies investigating the genetic basis of verbal memory in adolescents. It benefited greatly from the diversity within the ABCD Study^®^, which allowed our GWAS to identify two genome-wide significant associations for RAVLT short-delay recall and long-delay recall. Genetic correlations estimated by LDSC based on the European GWAS revealed negative genetic correlations between schizophrenia and the three verbal memory traits, providing further evidence that supports verbal memory as a psychosis endophenotype.

Our study included all participants in the ABCD Study^®^, which increased the representation of diverse populations and the power to detect genome-wide significant associations [65]. An alternative way to perform GWASs in diverse populations is to stratify participants into different ancestral groups, conduct the GWASs separately, and meta-analyse the results together. However, because of the large proportion of admixed individuals in the ABCD Study^®^, such stratification could cause substantial sample loss. Instead, we employed a mixed-model approach using the GENESIS package to analyse the whole sample altogether while adjusting for ancestral background and familial relatedness. We successfully distinguished individual relatedness due to family structure from their ancestral backgrounds. This allowed for better adjustment for ancestry, especially for SNPs with relatively small or large differences in allele frequency across ancestries, which are usually not fully controlled for by other mixed-model methods [37]. The inclusion of all participants also boosted the power of the GWAS and it is only with this larger sample we identified two genome-wide significant associations. The associations were not identified in the European subset, most probably due to having just half the sample size and the low allele frequency of rs73984566 in European populations (MAF < 0.001).

We found one locus located in chromosome 2 (lead SNP: rs73984566) associated with RAVLT short-delay recall. Chromatin interaction data provided evidence for two genes associated with this locus, NABP1 and TMEFF2. NABP1 (Nucleic Acid Binding Protein 1) is involved in many ubiquitous DNA metabolic processes [66], but its role in brain development remains unclear and needs further research. More information is available for the TMEFF2 (transmembrane protein with EGF-like and two follistatin-like domains 2) gene, which is widely expressed in the brain and has been proposed to be a survival factor for neurons in the hippocampus and midbrain [67]. The neuroprotective effect of TMEFF2 has also been found in Alzheimer’s disease as it binds Amyloid-β oligomer and Amyloid-β protein precursor [68]. One study found that methylation of the TMEFF2 promoter region was associated with poor outcomes in patients with glioma, a type of brain tumour [69]. However, few studies have investigated the role of TMEFF2 in a neurodevelopmental context, and more research is needed to clarify its role in adolescent verbal memory.

We also found RAVLT short-delay recall to be associated with the 17q21.31 locus. The lead SNP, rs9896243, is located in the intronic region of the NSF (N-ethylmaleimide sensitive factor, vesicle fusing ATPase) gene. eQTL data provided further evidence for the role of this gene. NSF is an important factor in synaptic neurotransmission that facilitates the recycling of SNARE (soluble N-ethylmaleimide sensitive factor attachment protein receptors) proteins, which mediate membrane fusion [70]. Previous GWASs have reported many associations between the NSF gene and brain volume or structure measures [71,72], Parkinson’s disease [73], and neuroticism [74]. Notably, the NSF gene has also been identified by previous GWASs on general cognitive ability [20,21], which is in line with the phenotypic correlation observed between general cognitive ability and verbal memory [26]. This suggests that the heritability of general cognitive ability could be further partitioned into specific cognitive domains, although unique genetic influences on specific cognitive domains may also exist. Moreover, the same SNP (rs9896243) was found to be associated with worry measurement at the genome-wide significance level in a GWAS on neuroticism [74], which is a risk factor for various psychiatric conditions including psychosis. Indeed, increased SNARE protein–protein interactions have also been found in postmortem brain samples of patients with schizophrenia, supporting the hypothesis of general synaptic dysfunction in schizophrenia [75]. Since adolescent verbal memory is an endophenotype for psychosis [14,15], our finding suggests that the NSF might play a role in the synaptic dysfunction in psychosis by modulating SNARE complex activity during neurodevelopment. The 17q21.31 locus is also associated with Koolen-de Vries syndrome [76,77,78], a disorder characterised by developmental delay and intellectual disability, thus providing further evidence for the involvement of this locus in neurodevelopment.

Based on the results of the GWAS in the European subset, we estimated the SNP heritability to be 15%, 29%, and 21% for RAVLT immediate recall, short-delay recall, and long-delay recall. This supports the influence of common variants on verbal memory in adolescents. A previous family study using a similar measure (the California Verbal Learning Test) estimated the heritability to be 35%, 50%, and 73% for the three verbal memory traits, respectively, [79]. The “missing heritability” estimated from the current GWAS is expected, given that more SNPs with small effects are yet to be found, as well as other factors that GWASs could not account for (e.g., rare variants) [80]. Moreover, heritability estimates could also be influenced by age, as the sample in Bertisch et al. included adults while the ABCD Study^®^ was only composed of children and adolescents [79]. There is evidence that the heritability of general cognitive ability increases during adolescence due to children interacting with their environments based on genetic propensities [14,15,16,17], although it is unclear if this pattern also exists for specific cognitive domains. Nevertheless, investigating the genetic basis of verbal memory during adolescence is still important, as it can help to identify relevant neurodevelopmental processes that manifest prior to the onset of many psychiatric conditions.

We found significant and positive genetic correlations between all three verbal memory traits and educational attainment, which is expected given the close relationship between the two phenotypes. We also found negative genetic correlations between all three verbal memory traits and schizophrenia, although the association was trend-level for RAVLT long-delay recall. However, as the genetic correlations within the three traits were very high and only European participants were included in this analysis, we believe the trend-level association will reach statistical significance with bigger sample sizes. This finding is in line with previous studies that found deficits in verbal memory amongst the relatives of patients with psychosis, supporting the role of verbal memory as a psychosis endophenotype [14,15,16,17]. Moreover, the negative genetic correlation between RAVLT short-delay recall and depression indicates that verbal memory deficits might not be specific to psychosis. Given the shared heritability within psychiatric disorders [81], verbal memory deficits in adolescence may represent general neurodevelopmental alterations that can manifest into different forms of psychiatric conditions later in life. It is worth noting that after correction by the number of external traits tested (new significance threshold: 0.05/4 = 0.0125), only the association between RAVLT immediate recall and schizophrenia remained significant. This could indicate that immediate verbal memory is a better psychosis endophenotype than the other two verbal memory traits. Thus, our findings should be viewed with caution and need to be replicated in the future.

The current study has limitations. Although benefiting from the large adolescent sample of the ABCD Study^®^, our GWAS is still underpowered to detect SNPs with small effect sizes. Due to the lack of public data available for adolescent verbal memory in diverse populations, we were also not able to conduct replication analysis in an independent sample. Although promising, our findings are preliminary, and future studies with diverse adolescent samples are still needed to replicate our findings. Moreover, as the current methods for estimating SNP heritability and genetic correlations requires LD references from a certain population, we were only able to perform those analyses in the European subset due to the limited sample sizes of other ancestries in the ABCD Study^®^. Methodological advances are needed in the future to allow for the inclusion of more diverse samples in such analyses. Furthermore, potential sampling bias might exist in the current GWAS. It is possible that children and adolescents with better general cognitive ability were more likely to participate in the ABCD Study^®^, causing the sample to be biased towards this demographic. Nonetheless, we believe that the study sample was representative of the general population at least in RAVLT performance, as the mean scores were comparable to previous results obtained from children in similar age groups [82,83]. Finally, we did not analyse chromosomal aberrations (e.g., copy number variants) in our study as this requires a different technique. As chromosomal aberrations have been associated with cognitive impairment [24,25], future studies should continue to investigate their influence on specific cognitive domains, such as verbal memory.

In summary, this GWAS, one of the first in children, identified two compelling loci associated with verbal memory that now warrant mechanistic research. We benefited from the diversity within the ABCD Study^®^ and conducted a joint analysis of participants from all ancestries, which maximised the statistical power enabling genome-wide significant findings. Notably, the NSF gene was identified by both SNP-based association testing and gene mapping, indicating that synaptic neurotransmission and membrane fusion might play an important role in verbal memory during neurodevelopment. We also found negative genetic correlations between verbal memory and schizophrenia, suggesting their shared underlying mechanisms. Future research employing novel approaches, including multi-trait analysis of GWAS summary statistics and pathway-specific polygenic risk scores [84,85], will be beneficial in revealing the shared biological pathways between verbal memory, psychosis, and other neuropsychiatric traits.

## Figures and Tables

**Figure 1 genes-13-00106-f001:**
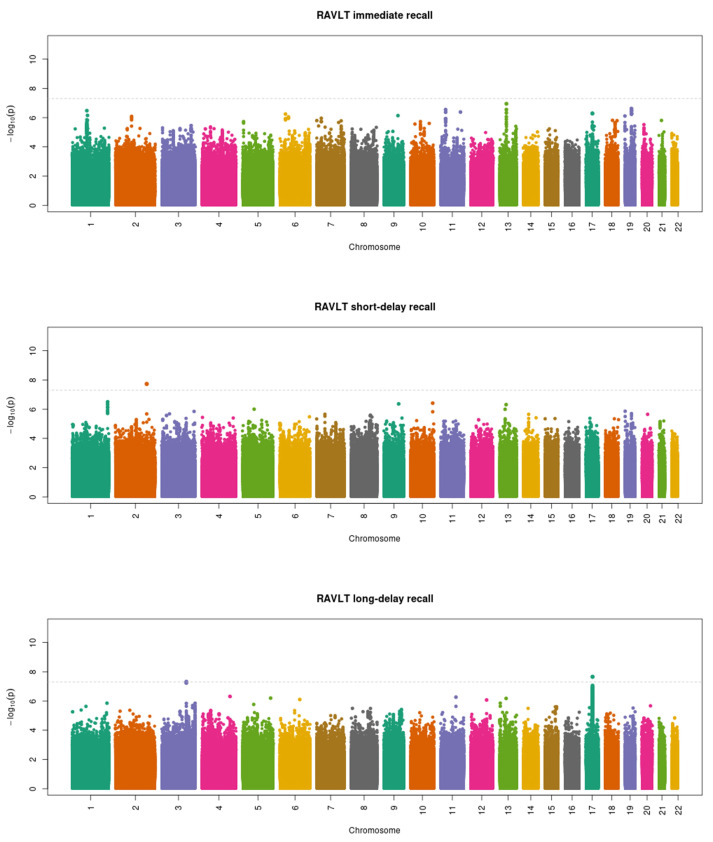
Manhattan plots of the genome-wide association analysis of the three verbal memory traits in the whole sample. The dotted line indicates the genome-wide significance threshold of *p* = 5 × 10^−8^. RAVLT, Rey Auditory Verbal Learning Test.

**Table 1 genes-13-00106-t001:** Demographic and cognitive characteristics of the Adolescent Brain Cognitive Development sample.

Variable	*n* (11,017 in Total)	%
**Sex**		
Female	5176	47
Male	5805	53
**Mean (SD) Age (years)**	9.9	0.6
**Ethnicity**		
White	5872	53
Hispanic	2099	19
Black	1684	15
Asian	155	1
AIAN/NHPI and mixed	1169	11
**Mean (SD) RAVLT score**		
Immediate recall	44.2	9.9
Short-delay recall	9.7	3
Long-delay recall	9.2	3.2

Note. There were 36 missing data for sex and age, 38 for ethnicity, 291 for RAVLT immediate recall, 216 for RAVLT short-delay recall, and 260 for RAVLT long-delay recall. SD, standard deviation. AIAN, American Indian/Alaska Native. NHPI, Native Hawaiian and other Pacific Islander. RAVLT, Rey Auditory Verbal Learning Test.

**Table 2 genes-13-00106-t002:** Genomic loci associated with verbal memory.

Phenotype	Lead SNP	Chromosome (Position)	Reference Allele	Effect Allele (Frequency)	Beta (SE)	*p*	*r* ^2^
RAVLT short-delay recall	rs73984566	2 (191566282)	G	A (0.019)	−0.847 (0.151)	1.86 × 10^−8^	0.91
RAVLT long-delay recall	rs9896243	17 (46748690)	C	G (0.185)	−0.309 (0.055)	2.22 × 10^−8^	0.96

Note. Genomic positions are based on GRCh38. Frequency represents allele frequency in the sample. RAVLT, Rey Auditory Verbal Learning Test. SE, standard error. *r*^2^, imputation quality score.

**Table 3 genes-13-00106-t003:** Genetic correlations between three verbal memory traits and relevant psychiatric traits or educational attainment.

Verbal Memory Traits	Schizophrenia		Bipolar Disorder		Major Depressive Disorder		Educational Attainment	
	*r_g_* (SE)	*p*	*r_g_* (SE)	*p*	*r_g_* (SE)	*p*	*r_g_* (SE)	*p*
RAVLT immediate recall	−0.29 (0.10)	0.003	−0.18 (0.10)	0.075	−0.14 (0.09)	0.092	0.53 (0.14)	0.001
RAVLT short-delay recall	−0.15 (0.06)	0.017	−0.11 (0.06)	0.082	−0.14 (0.07)	0.050	0.31 (0.06)	<0.001
RAVLT long-delay recall	−0.15 (0.08)	0.063	−0.13 (0.08)	0.083	−0.07 (0.08)	0.370	0.40 (0.08)	<0.001

Note. RAVLT, Rey Auditory Verbal Learning Test. *r_g_*, genetic correlation. SE, standard error.

## Data Availability

The data presented in this study are openly available in the ABCD Study^®^ at doi:10.15154/1524269. Summary statistics can be provided by the authors upon request.

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
