# Peer review of "Adolescent Verbal Memory as a Psychosis Endophenotype: A Genome-Wide Association Study in an Ancestrally Diverse Sample"

_genes, 2022, doi:10.3390/genes13010106_

Round 1

Reviewer 1 Report

Wang et al present a GWAS in multiple ancestries, examining verbal memory in a US adolescent population cohort. They use cutting-edge methods to include all participants in their analyses, addressing an important gap in the literature for multiple ancestry studies, and contributing to a relatively sparse literature on the genomics of verbal memory.

The paper has been well-written and well performed. I have several comments, which I include in order of appearance below:

  1. Line 37: the Cohen's d for verbal memory in bipolar disorder is cited as -0.56 to -0.56, which seems highly precise. I could not entirely determine where in the reference this figure comes from - the authors should review to check this isn't a typo.
  2. Line 37: The phrase "Similar yet milder" seems self-contradictory - "Milder" alone would suffice.
  3. Line 40: Deficits in relatives of patients are consistent with a genetic effect, but do not themselves suggest such an effect - for example, you would see the same thing with a shared environmental aetiology. The authors might consider rephrasing "Those findings suggest that verbal memory impairment is associated with the genetic risk of psychosis" to "Those findings are consistent with verbal memory impairment being associated with the genetic risk of psychosis"
  4. Line 46: Although GWAS in verbal memory are relatively sparse, the broader construct of cognitive ability is a very well-studied GWAS phenotype, including in children. It would be informative if the authors briefly mentioned the relationship between verbal memory and cognitive ability and highlight this literature, and they should also discuss their findings in the context of previous findings in cognitive ability.
  5. Line 75: The need for homogeneity is not really the issue with the stratified meta-analysis approach, but rather that sufficient sample sizes are needed in each homogenous ancestral group to estimate the small effects of individual variants with sufficient precision.
  6. Line 76: The authors need to justify why joint mixed models might be the optimal solution to the problem here (at least briefly and at a high-level, as could be achieved by summarising the argument from the reference).
  7. Line 100: Please can the authors briefly summarise the recruitment procedures of the ABCD study (in a Supplement if necessary)?
  8. Line 102 (and more general): It would be informative to provide information on the distributions of the phenotypes used for GWAS - do these conform to the assumptions of the mixed linear model (e.g. https://besjournals.onlinelibrary.wiley.com/doi/10.1111/2041-210X.13434)?
  9. Line 119: please provide details of the quality control procedures (e.g. parameters examined, cut-offs used) conducted by ABCD in the Supplement.
  10. Line 124: The approach described here effectively removes any uncertainty from the imputation of the variants - genotype identifier is assigned to the most likely call (even if that is only trivially more likely than the next most likely call), and variants with relatively little information content (equivalent to 30% of the total N) are included. Neither of these things is a major problem (although the hard call threshold is lower than is typical), but the top SNPs should be checked for their INFO score (which should be reported), and nearby genotyped SNPs should be checked for their association with the phenotypes to ensure it is consistent with imputed lead SNPs (and therefore that it is unlikely that significant associations with imputed SNPs are due to any imputation biases). If possible, repeating analyses using the dosage data for the two lead SNPs would be a useful statistical validation (but I am aware that may require using a different analytical method, and so introduce too many variables for an inconsistent result to be interpretable)
  11. Line 152: Please can the authors justify the choice of 4 or 8 PCs?
  12. Line 159 (and more general): Please can the authors justify their definition of locus boundaries? Given that the r2 between two variants can vary between ancestral populations, and that the 1KG reference panel will be comprised of a different mixture of ancestral groups to the ABCD study (and so have a different mean average LD structure), how reliable is this method for defining locus boundaries? Would it be preferable to define these boundaries using the data itself (i.e. in PLINK?)
  13. Line 205: It is not clear why it is necessary to refer to the group of Native Hawaiian, Pacific Islander, Alaskan Native, Native American and mixed ethnicity as "Other".  This composite group is never referred to specifically again in the paper. The authors should just list these ethnicities in the table (as a composite if necessary for e.g. privacy reasons - but the "Other" label is unnecessary and somewhat exclusionary).
  14. Table 3: Results are reported without consideration for multiple testing. Given there are four external disorders/traits being examined, it would be reasonable to set significance at p<0.0125, which changes the interpretation of the results. At the least, the authors should be more skeptical of the significance of an association with p = 0.05.
  15. Line 368: The three phenotypes are highly correlated - as such, the fact that lead variants from one phenotype have consistent associations with the other phenotypes is not surprising and does not really increase the reliability nor the robustness of the findings in my view. I would remove this sentence. The caution the authors otherwise use in this paragraph is warranted - their results are at the edge of significance in what is a small (given the likely effect sizes of variants acting on verbal memory) and  heterogeneous sample, which (for valid reasons) does not have a suitable replication cohort. The results are promising but preliminary - that's fine, and the authors should not be afraid to be explicit about that.

Reviewer 2 Report

Wang et al. investigated the genetic basis of verbal memory in a neurodevelopmental context. There were not so many similar studies in current literature, especially conducted in other than non-European populations. The study is well designed. Two novel genome-wide significant associations were found for short-delay and long-delay recall verbal memory. The paper is well-written. However I have some minor comments for authors:

-Did any of the participants of the study have any comorbidity that might potentially affect the verbal memory?

-In my opinion authors should discuss in the introduction section that there are some genetic variation like chromosomal aberrations which may also potentially affect the cognition (ie. Ropers HH. Annu Rev Genomics Hum Genet. 2010;11:161-87., Figura et al. Neurol Neurochir Pol 2021;55(3):300-305.)
